# Multiyne chains chelating osmium via three metal-carbon σ bonds

Qingde Zhuo[1], Jianfeng Lin[1], Yuhui Hua[1], Xiaoxi Zhou[1], Yifan Shao[1], Shiyan Chen[1], Zhixin Chen[1], Jun Zhu [1], Hong Zhang[1] & Haiping Xia [1]

Although the formation of metal–carbon σ bonds is a fundamental principle in organometallic chemistry, the direct bonding of one organic molecule with one metal center to generate more than two metal–carbon σ bonds remains a challenge. Herein, we report an aromaticity-driven method whereby multiyne chains are used to construct three metal–carbon σ bonds in a one-pot reaction under mild conditions. In this method, multiyne chains act as ligand precursors capable of chelating an osmium center to yield planar metallapolycycles, which exhibit aromaticity and good stability. The direct assembly of various multiyne chains with commercially available metal complexes or even simple metal salts provides a convenient and efficient strategy for constructing all carbon-ligated chelates on the gram scale.

[1] State Key Laboratory of Physical Chemistry of Solid Surfaces and Collaborative Innovation Center of Chemistry for Energy Materials (iChEM), College of Chemistry and Chemical Engineering, Xiamen University, Xiamen 361005, China. Correspondence and requests for materials should be addressed to H.X. (email: hpxia@xmu.edu.cn)

The construction of metal–carbon σ bonds is a fundamental process in organometallic chemistry[1,2]. A large array of organic molecules form a single σ bond with metals, resulting in species such as metal alkyl[3,4], metal carbene[5–12], and metal carbyne[13–18] complexes. Certain organic molecules can bond with a metal center through two metal–carbon σ bonds, leading to the corresponding bidentate CC-type chelates, such as metallacyclopentadienes[19–25] and metallabenzenes[26–34]. However, the direct σ-bonding of a single organic molecule to a single metal center to form tridentate CCC-type chelates is uncommon. Most of them contain N-heterocyclic carbene ligands, which are electron-rich σ-donor ligands that can strongly bond the metal centers in a manner similar to classical 2e donors such as phosphine ligands[35–37]. Another related example is the tantalum complexes derived from 1-iodo-2,6-di-p-tolylbenzene through lithiation and cyclometallation processes, in which the terphenyl ligand attaches to a metal center through three metal–carbon single bonds[38].

In recent years, our research has focused on the construction of as many metal–carbon σ bonds as possible between a carbon ligand and a metal center, achieving a series of conjugated carbon-ligated multidentate chelates, termed carbolong complexes, with novel structures and unique properties[39–43]. All of these complexes contain three or more metal–carbon σ bonds; however, these metal–carbon σ bonds could only be constructed step by step.

Aromaticity-driven cyclization reactions have been widely used in synthetic chemistry owing to the relatively lower energies of the aromatic products compared to their non-aromatic or anti-aromatic analogs. Numerous aromatic hydrocarbons have been constructed using this method. As shown in Fig. 1a, one of the most prominent examples of this chemistry is the transition metal-catalyzed [2 + 2 + 2] cycloaddition reactions of alkynes, which can form three carbon–carbon σ bonds in a one-pot reaction to obtain benzene and its derivatives[44]. Thus, we considered whether the direct construction of three metal–carbon σ bonds could also be achieved through an aromaticity-driven process. Herein, we present an efficient strategy involving aromaticity-driven processes in a one-pot reaction to achieve stable multidentate complexes containing three metal–carbon σ bonds (Fig. 1b). The direct reaction of multiyne chains with a commercially available metal complex ($OsCl_2(PPh_3)_3$) or even a simple metal salt ($K_2OsCl_6$) affords all carbon-ligated chelates with both single and multiple metal–carbon bonds under ambient conditions on the gram scale. The promising synthetic potential of this method is evidenced by the extension of the multiynes to produce diverse chelate scaffolds.

## Results

**Synthesis of CCC-type chelates with three M–C σ bonds.** We first designed and prepared the carbon ligand precursor **L1** (Fig. 2a; see Supplementary Methods for synthesis details). **L1** is a triyne chain containing two functional modules (an acetenyl unit and a 3-hydroxyl pentadiynyl unit) connected by a flexible linker. As shown in Fig. 2a, the treatment of **L1** with $OsCl_2(PPh_3)_3$ and $PPh_3$ in dichloromethane under air at room temperature (RT) for 1 h resulted in the formation of CCC-type chelate **1** with a yield of 82%.

Chelate **1** was characterized by nuclear magnetic resonance (NMR) spectroscopy. The $^{13}C$ NMR spectrum shows a typical chemical shift of the carbyne carbon in **1** at 321.57 parts per million (ppm). The downfield $^{1}H$ NMR chemical shifts (H7 at 13.14 ppm and H3 at 7.64 ppm) appear to be diagnostic of the presence of an aromatic framework, which is common for aromatic metallacycles[26–34,45–47]. Single-crystal X-ray analyses indicated that a tridentate CCC-type ligand was attached to the osmium center by three metal–carbon σ bonds (Fig. 2b), confirming that the combination of **L1** with the osmium substrate led to the conversion of five sp carbons and one $sp^3$ carbon to six $sp^2$ carbons. The Os–C1 bond length (1.840(7) Å, where the value in parentheses represents the standard deviation) is at the high end of the range for Os≡C triple bonds (1.612–1.856 Å). However, the other two Os–C bond lengths (i.e., 2.108(7) Å for Os–C4 and 2.040(8) Å for Os–C7) are within the range of the

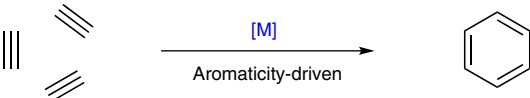

**a** Aromaticity-driven formation of three C–C σ bonds

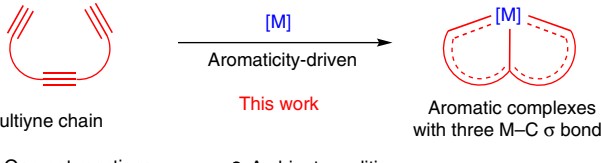

**b** Aromaticity-driven formation of three M–C σ bonds

Multiyne chain This work Aromatic complexes with three M–C σ bonds

- One-pot reactions
- Ambient conditions
- Scalable synthesis
- Commercially available metal sources

**Fig. 1** Efficient synthesis through aromaticity-driven cyclization reactions. **a** Formation of three C–C σ bonds through an aromaticity-driven process. **b** Formation of three M–C σ bonds through an aromaticity-driven process. [M] transition metal and ancillary ligands, M–C metal–carbon

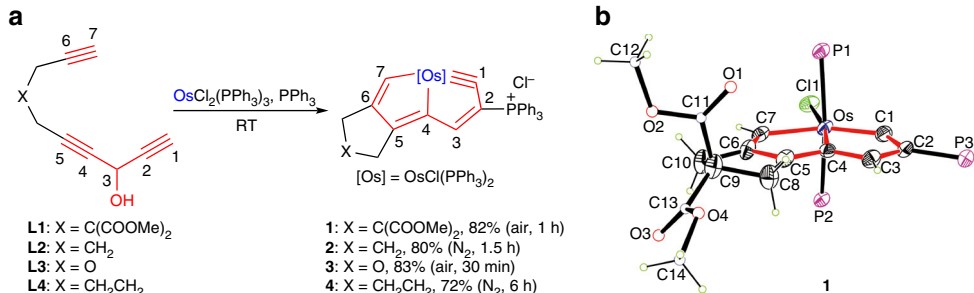

L1: X = C(COOMe)₂ 1: X = C(COOMe)₂, 82% (air, 1 h)
L2: X = CH₂ 2: X = CH₂, 80% (N₂, 1.5 h)
L3: X = O 3: X = O, 83% (air, 30 min)
L4: X = CH₂CH₂ 4: X = CH₂CH₂, 72% (N₂, 6 h)

[Os] = OsCl(PPh₃)₂

**Fig. 2** Synthesis of CCC-type osmium chelates from triynes. **a** Triynes as ligand precursors to access CCC-type chelates with three metal–carbon σ bonds. **b** X-ray structure of the cation of chelate **1** drawn with 50% probability (phenyl groups in PPh₃ are omitted for clarity). Selected bond distances (Å) and angles (deg): Os–C1 1.840(7), Os–C4 2.108(7), Os–C7 2.040(8), C1–C2 1.406(10), C2–C3 1.394(10), C3–C4 1.395(9), C4–C5 1.390(10), C5–C6 1.395(10), C6–C7 1.366(10), Os–C1–C2 129.3(6), C1–C2–C3 108.3(6), C2–C3–C4 112.1(6), C3–C4–Os 117.1(5), C1–Os–C4 73.2(3), Os–C4–C5 116.6(5), C4–C5–C6 113.6(7), C5–C6–C7 115.5(7), C6–C7–Os 118.9(5), C7–Os–C4 75.4(3)

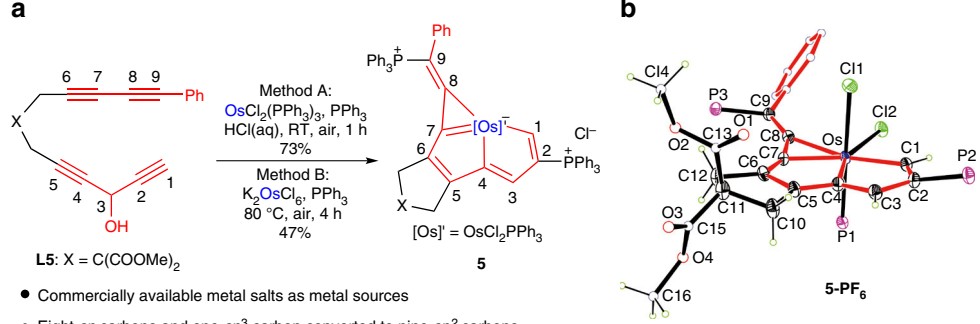

**Fig. 3** Synthesis of a CCCC-type chelate from tetrayne and alternative metal sources. **a** Tetrayne as ligand precursor to access a CCCC-type chelate. **b** X-ray structure of the cation of **5-PF₆** drawn with 50% probability (phenyl groups in PPh₃ are omitted for clarity). Selected bond distances (Å) and bond angles (deg): Os–C1 2.041(3), Os–C4 2.082(3), Os–C7 2.003(3), Os–C8 2.132(3), C1–C2 1.385(5), C2–C3 1.413(5), C3–C4 1.378(5), C4–C5 1.416(5), C5–C6 1.379(5), C6–C7 1.401(5), C7–C8 1.361(5), C8–C9 1.350(5), Os–C1–C2 118.7(2), C1–C2–C3 114.6(3), C2–C3–C4 112.7(3), C3–C4–Os 118.6(2), C1–Os–C4 75.36(13), Os–C4–C5 118.0(2), C4–C5–C6 113.9(3), C5–C6–C7 111.0(3), C6–C7–Os 123.2(2), C7–Os–C4 73.85(13), Os–C7–C8 76.0(2), C7–C8–Os 65.7(2), C8–Os–C7 38.26(13), C7–C8–C9 143.2(3)

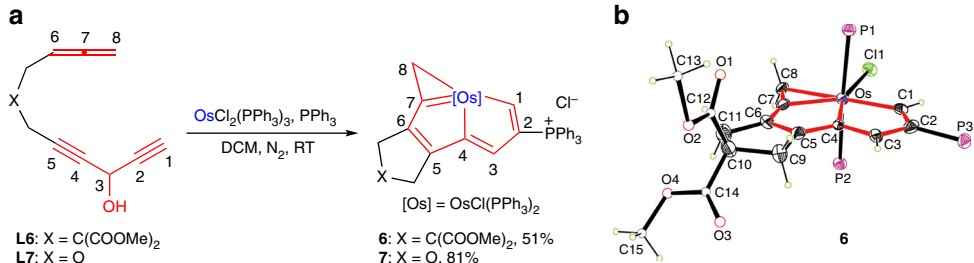

**Fig. 4** Synthesis of CCCC-type chelates from allene group substituted diynes. **a** Allene group substituted diynes as ligand precursors to access CCCC-type chelates. **b** X-ray structure of the cation of **6** drawn with 50% probability (phenyl groups in PPh₃ are omitted for clarity). Selected bond distances (Å) and bond angles (deg): Os–C1 2.017(7), Os–C4 2.074(7), Os–C7 2.015(8), Os–C8 2.270(7), C1–C2 1.383(11), C2–C3 1.403(11), C3–C4 1.381(10), C4–C5 1.395(11), C5–C6 1.385(10), C6–C7 1.349(12), C7–C8 1.403(11), Os–C1–C2 119.1(6), C1–C2–C3 114.4(7), C2–C3–C4 112.8(7), C3–C4–Os 118.0(6), C1–Os–C4 75.7(3), Os–C4–C5 117.9(5), C4–C5–C6 113.8(7), C5–C6–C7 111.4(7), C6–C7–Os 123.4(6), C7–Os–C4 73.4(3), Os–C7–C8 81.1(5), C7–C8–Os 61.3(4), C8–Os–C7 37.6(3)

Os–C single bonds (1.921–2.195 Å) of vinyl osmium complexes (these ranges are based on a search of the Cambridge Structural Database, CSD version 5.38, in November 2016). The carbon–carbon bond distances within the metallacycles (1.366 (10)–1.406(10) Å) lie between the typical single and double carbon–carbon bond distances. The metallabicycle is nearly coplanar, as reflected by the small mean deviation from the least-squares plane (0.020 Å). The planarity of the metallacycles and the carbon–carbon bond distance equivalency indicate extensive electronic delocalization within the osmabicycle ring in **1**. All the salient features of **1** compare well with those of previously reported aromatic osmapentalynes[39]. Osmapentalynes and related osmium chelates with multidentate carbon ligands have attracted considerable attention as novel aromatic molecules with unique properties, especially in the area of photophysics[39–43, 48]. However, previous methods reported for the synthesis of these chelates with multidentate carbon ligands require the use of an unstable osmacycle precursor via standard Schlenk techniques[39–43, 48]. By contrast, triyne **L1** can lead to an osmapentalyne in a one-pot synthesis under an atmosphere of air and from commercially available metal source.

The facile synthesis of **1** using this method demonstrates that it is highly efficient for constructing complexes with three metal–carbon σ bonds via the cyclization of multiyne chains with metal species. To explore the generality of our strategy, we prepared multiynes with different linkers. Similar to **L1**, **L2–L4** were able to chelate an osmium center via three metal–carbon σ bonds in a one-pot reaction to achieve CCC-type chelates **2–**

**4**, respectively, in high yield (Fig. 2a). These results demonstrate that our method can be applied to a broad range of substrates.

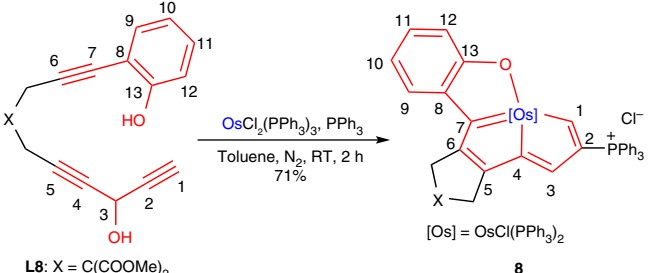

**Fig. 5** Synthesis of CCCO-type chelate from a phenolic group substituted triyne. Phenolic group substituted triyne as ligand precursor to access CCCO-type chelate

**Extension of the multiyne platform**. The efficient ligating ability of **L1–L4** inspired us to investigate whether an extension of the multiyne chain will provide tetradentate carbon-ligated chelates and, if so, how the synthesis is achieved.

As shown in Fig. 3a, the reaction of tetrayne **L5** with OsCl₂(PPh₃)₃ in air afforded complex **5** with a yield of 73%. The crystal structure of **5-PF₆** (the derivative of **5** with PF₆⁻ as the counteranion) reveals that **5** has a fused metallapentalene

structure with the metallacyclopropenylidene moiety, which is a unique planar tetradentate CCCC-type chelate (Fig. 3b). The metallatricycle is basically coplanar, as reflected by the small mean deviation from the least-squares plane (0.076 Å). The osmium–carbon bond distances within the five-membered metallabicycle (2.041(3) Å for Os–C1, 2.082(3) Å for Os–C4, and 2.003(3) Å for Os–C7) are in agreement with those of osmapentalenes (1.926(6)–2.139(6) Å) reported previously[49], whereas the Os–C8 length is 2.132(3) Å, which is comparable to those of osmium vinyl complexes (1.921–2.195 Å). C8–C9 (1.350(5) Å) is a typical exocyclic double bond. By contrast, the C–C bond distances within the tricyclic skeleton are all between the carbon–carbon single- and double-bond distances, suggesting that there is π-electron delocalization within the fused metalla-cycle (see Supplementary Fig. 1 for resonance structures). Consistent with the experimental observations, theoretical calculations (natural bond orbital analyses) also indicated that 5 is a tetradentate CCCC-type chelate (Supplementary Fig. 2). It is worth noting that the eight C($sp$) and one C($sp^3$) atoms of L5 were completely converted into nine C($sp^2$) atoms and were incorporated into the final planar conjugated skeleton.

Interestingly, the formation of 5 was also achieved by reacting L5 with the inorganic salt $K_2OsCl_6$ under air (Fig. 3a). Although the reactions of triynes L1–L4 with inorganic salts did not work, the reaction of tetrayne L5 with $K_2OsCl_6$ indicates that the direct combination of inorganic salts with multiynes could be a readily accessible strategy. This approach may provide opportunities for the use of multiyne chains as carbon ligand precursors; that is extending multiyne chains could enhance their chelating ability, which would lead to the synthesis of useful chelates in an efficient manner.

Multiyne chains with allene groups were also compatible with our strategy. As shown in Fig. 4a, the reaction of L6 (or L7) with $OsCl_2(PPh_3)_3$ led to the formation of complex 6 (or 7) with a yield of 51% (or 81%). X-ray crystallography analyses confirmed that both 6 (Fig. 4b) and 7 (Supplementary Fig. 6) were tetradentate CCCC-type chelates. The main difference between 5 and 6/7 is the hybrid form of C8, which is $sp^2$-hybridized in 5 but $sp^3$-hybridized in 6 and 7.

In addition to all carbon-ligated chelates, chelates with hetero coordination atoms can also be constructed via the cyclization reactions of multiyne chains with metals. As shown in Fig. 5, the reaction of the phenolic group substituted triyne L8 with $OsCl_2(PPh_3)_3$ afforded the tetradentate CCCO-type chelate 8 with a yield of 71%. 8 was characterized by NMR spectroscopy and all of its characteristic chemical shifts of protons, carbons, and phosphorus were consistent with those of recently reported values for osmapentalenofuran[50]. The structure of 8 was further confirmed by high-resolution mass spectrometry (HRMS), and

elemental analyses. It is noteworthy that 8 represents a rare example of a planar tetradentate CCCO-type chelate[50].

**DFT studies of the aromaticity of 1–8.** Consistent with their structural aromatic characteristics, 1–8 exhibit good stability (all of them stable in air at RT). Density functional theory (DFT) calculations were performed to evaluate their aromaticity. Taking 1 as an example, the calculated aromatic stabilization energy (ASE) results are comparable to the values reported for other metallaaromatics (Fig. 6a)[39, 49–54]. We also calculated the nucleus-independent chemical shift (NICS) along the z-axis at 1 Å above the ring critical point (NICS(1)$_{zz}$; the average value was used when the environments above and below the ring centers were not equivalent) of model compound 1′, which we simplified by replacing the PPh$_3$ groups with PH$_3$ groups (Fig. 6b)[55–57]. Both the considerable ASE value and negative NICS values suggest that the ring systems in complex 1 are aromatic. The anisotropy of the induced current density (AICD) method[58, 59] was applied to visualize the electron delocalization within the metallacycles (Fig. 6c). The aromatic character was further confirmed by the obvious diatropic ring current (clockwise vectors) within the metallabicyclic moiety of model osmapentalyne 1′. The NICS(1)$_{zz}$ and ASE values (Table 1; see Supplementary Figs. 8 and 9 for details), and AICD analyses (Supplementary Figs. 11–17) of complexes 2–8 are also in good agreement with their aromatic natures. Thus, we propose that the aromaticity of the final products is crucial for the assembly of multiynes with osmium in a one-pot reaction.

## Discussion

To characterize the aromaticity-driven processes for multiynes chelating osmium, we performed DFT calculations. The computed free-energy profile of the key steps in the reaction of L1 with the osmium center to yield the final σ-bonded CCC-type chelate 1 is shown in Fig. 7. The multiyne L1 may initially bind to the metal to form intermediate A1 through the dissociation of the

**Table 1 Aromaticity evaluations of the cation moieties of complexes 1-8**

| Complex | NICS(1)zz (ppm) (A/B) | Complex | ASE (kcal/mol) |
|---|---|---|---|
| 1′+ | –17.3/–14.3 | 1+ | 29.5 |
| 2′+ | –17.6/–14.1 | 2+ | 26.7 |
| 3′+ | –17.7/–14.3 | 3+ | 14.6 |
| 4′+ | –17.2/–15.2 | 4+ | 20.8 |
| 5′+ | –13.6/–8.0 | 5+ | 25.5 |
| 6′+ | –16.0/–6.8 | 6+ | 25.8 |
| 7′+ | –16.2/–6.7 | 7+ | 14.1 |
| 8′+ | –17.5/–5.3 | 8+ | 21.7 |

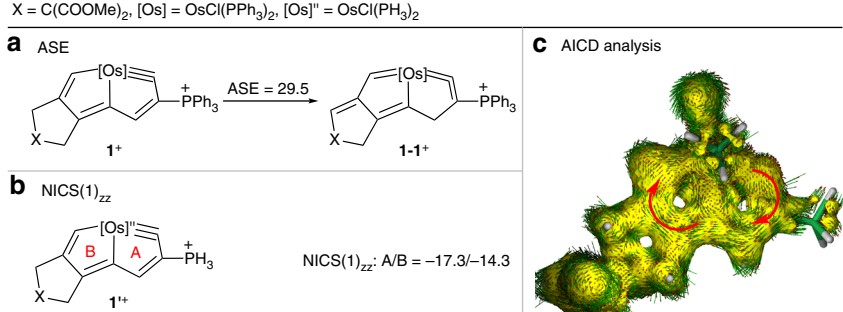

**Fig. 6** Aromaticity of the cation moiety of complex 1. **a** Aromatic stabilization energy (ASE, kcal/mol) evaluations of the aromaticity of the cation moiety of complex 1. **b** NICS(1)$_{zz}$ (nucleus-independent chemical shift along the z-axis at 1 Å above the ring critical point, ppm) evaluations of the aromaticity of the cation moiety of model complex 1′. **c** Anisotropy of the induced current density (AICD) plot of the cation moiety of model complex 1′ with an isosurface value of 0.025. The magnetic field vector is orthogonal to the ring plane and points upward (aromatic species exhibit clockwise diatropic circulations)

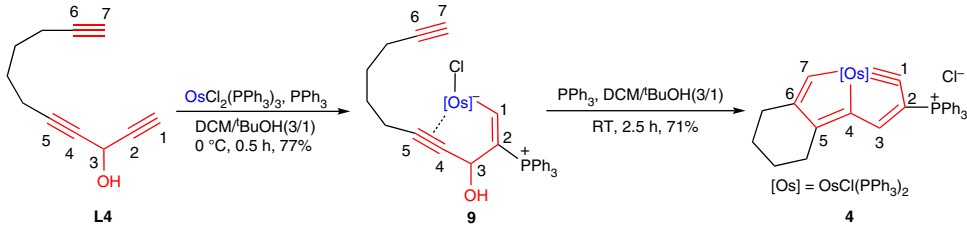

**Fig. 7** Gibbs free energy profile for the DFT-calculated formation mechanism of complex **1** at 298 K. The computed free energies are in kcal/mol

**Fig. 8** Isolation of the key intermediate. Isolation of the alkynyl-coordinated osmium vinyl intermediate **9** and its transformation to CCC-type osmium chelate **4**

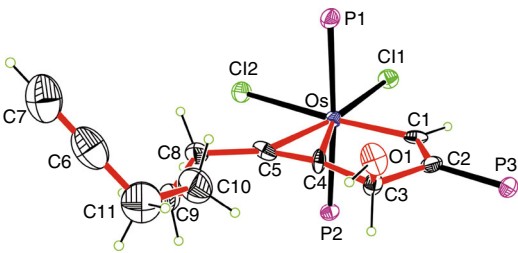

**Fig. 9** X-ray molecular structure of complex **9** drawn with 50% probability. Phenyl groups in PPh₃ are omitted for clarity. Selected bond distances (Å) and bond angles (deg): Os-C1 2.019(7), Os-C4 2.153(7), Os-C5 2.220(7), C1-C2 1.373(11), C2-C3 1.524(11), C3-C4 1.480(10), C3-O1 1.397(10), C4-C5 1.248(12), C5-C8 1.473(12), C8-C9 1.546(12), C9-C10 1.532(16), C10-C11 1.565(16), C6-C11 1.487(19), C6-C7 1.28(2), Os-C1-C2 126.0(6), C1-C2-C3 114.4(6), C2-C3-C4 104.6(6), C2-C3-O1 112.4(7), C3-C4-Os 122.4(5), C1-Os-C4 72.6(3), Os-C4-C5 76.4(5), C4-C5-Os 70.5(4), C4-Os-C5 33.1(3), Os-C5-C8 135.4(6), C11-C6-C7 177.6(14)

PPh₃ ligand from the metal center of OsCl₂(PPh₃)₃, which was computationally found to be facile. The following nucleophilic addition of PPh₃ to the terminal coordinated acetenyl group leads to the formation of the first metal–carbon σ bond, which could generate the alkynyl-coordinated osmium vinyl intermediate **B1**. This process has an energy barrier of 13.0 kcal/mol and is exergonic by 29.3 kcal/mol. The formation of the other two metal–carbon σ bonds occurs in two steps: the coordination of the remaining terminal C≡C bond with the metal center to form **C1** and [2 + 2 + 1] cyclization to furnish the bicyclic intermediate **D1** (with an energy barrier of 23.1 kcal/mol and exergonic of 30.6

kcal/mol from **B1** to **D1**). Finally, an acid-mediated aromatization can yield the final energetically favorable aromatic product **1** with three metal–carbon σ bonds, which is substantially more stable (by 21.2 kcal/mol) than **E1**. DFT calculations clearly show that the aromatic product **1** is more stable (more than 17 kcal/mol) than all the intermediates, which provides a thermodynamic driving force for the formation of three metal–carbon σ bonds.

The rationality of the aromaticity-driven mechanism was further verified by the isolation of a key intermediate for the synthesis of CCC-type chelate **4**. As shown in Fig. 8, when the reaction of **L4** with OsCl₂(PPh₃)₃ was carried out in a mixed solvent of dichloromethane and tert-butanol (3/1, v/v) at 0 °C for 0.5 h, complex **9** was isolated with a yield of 77%. Although **9** is labile in common organic solvents, it can be stable in tetrahydrofuran for weeks. Single-crystal X-ray diffraction (Fig. 9) reveals that **9** is an alkynyl-coordinated osmium vinyl complex with a free terminal C≡C bond, which is consistent with the intermediate **B1** shown in Fig. 7. Other related metal vinyl complexes have also been isolated as intermediates that were demonstrated to be highly reactive[60, 61]. Complex **9** could convert to osmapentalyne **4** when the reaction temperature was increased to RT. As monitored by in situ NMR, the conversion of **L4** to intermediate **9** and finally to chelate **4** further confirms that **9** is a key intermediate in the aromaticity-driven cyclization reaction (see Supplementary Fig. 21 for in situ ³¹P NMR monitoring spectra). Attempts to capture the corresponding intermediates related to **C1** and **D1** were unsuccessful, probably due to the strong aromaticity driving force of the final product.

In summary, we developed a highly efficient multidentate carbon ligand platform that involves the use of multiyne chains for the preparation of metal chelates with three or more metal–carbon σ

bonds in a one-pot reaction under mild conditions. The multiyne chains consist of two functional modules and a linker, we term which carbolongs. They can be used to react with a commercially available metal complex, or even a simple metal salt, affording all carbon-ligated chelates on the gram scale. Our theoretical and experimental studies revealed that aromaticity-driven processes play a substantial role in the high efficiency and robust characteristics of chelating metals with these multiyne chains. The application of this strategy to utilize various multiynes as a carbon ligand platform will potentially gain widespread use as a promising complement to existing techniques for the preparation of polycyclic organometallic complexes.

## Methods

**General methods**. Details of the synthesis and characterization of the multiynes and transition metal complexes can be found in the Supplementary Information. For X-ray data of complexes **1**, **5-PF$_6$**, **6**, **7**, and **9**, see Supplementary Tables 2, 3. For the detailed crystal structures of complexes **1**, **5-PF$_6$**, **6**, **7**, and **9**, see Supplementary Figs. 3–7. For the mechanisms of the formation of complexes **5**–**8**, see Supplementary Figs. 18–20. For the HRMS, $^1$H, $^{31}$P NMR, and $^{13}$C NMR spectra of the complexes in this article, see Supplementary Figs. 22–97. For the thermal decomposition data of complexes **1**–**8**, see Supplementary Table 1. The synthesis procedures for **1**, **5**, **7**, and **8** are representative of those for the preparation of all the described transition metal complexes.

**Synthesis of tridentate CCC-type chelate 1**. A dichloromethane solution (5 mL) of **L1** (1.20 g, 4.58 mmol) was added slowly to a green solution of OsCl$_2$(PPh$_3$)$_3$ (4.01 g, 3.83 mmol) and PPh$_3$ (5.02 g, 19.1 mmol) in dichloromethane (150 mL) under an air atmosphere, and the reaction mixture was stirred at RT for 1 h to yield a brown solution. The solution was evaporated under vacuum to a volume of ~15 mL and then washed with Et$_2$O (3 × 200 mL) to afford a brown solid. The solid was purified by flash chromatography on silica gel (eluent: 20:1 dichloromethane/ methanol) to yield complex **1** as a brown solid. Yield: 4.05 g, 82%.

**Synthesis of tetradentate CCCC-type chelate 5**. HCl (1.30 M aqueous solution 146 μL, 0.190 mmol) and a chloroform solution (2 mL) of **L5** (70 mg, 0.193 mmol) were added to a green solution of OsCl$_2$(PPh$_3$)$_3$ (200 mg, 0.191 mmol) and PPh$_3$ (250 mg, 0.953 mmol) in chloroform (10 mL) under an air atmosphere, and the reaction mixture was stirred at RT for 1 h to yield a red solution. The solvent was completely removed under vacuum. The resulting residue was dissolved in chloroform (2 mL) and washed with Et$_2$O (3 × 20 mL) to afford a red solid, and the solid was purified by column chromatography on silica gel (eluent: 20:1 dichloromethane/methanol) to yield complex **5** as a red solid. Yield: 199 mg, 73%. Please see Supplementary Methods for an alternative method for synthesis of complex **5**.

**Synthesis of tetradentate CCCC-type chelate 7**. A dichloromethane solution (2 mL) of **L7** (130 mg, 0.802 mmol) was added slowly to a green solution of OsCl$_2$(PPh$_3$)$_3$ (550 mg, 0.525 mmol) and PPh$_3$ (692 mg, 2.64 mmol) in dichloromethane (20 mL). The reaction mixture was stirred at RT for 30 min to yield a brown solution. The solution was evaporated under vacuum to a volume of ~2 mL and washed with Et$_2$O (3 × 30 mL) to afford a brown solid. The solid was purified by flash chromatography on silica gel (eluent: 20:1 dichloromethane/methanol) to yield complex **7** as a yellow solid. Yield: 506 mg, 81%.

**Synthesis of tetradentate CCCO-type chelate 8**. A toluene solution (5 mL) of **L8** (570 mg, 1.61 mmol) was added in one portion to a green solution of OsCl$_2$(PPh$_3$)$_3$ (1.36 g, 1.30 mmol) and PPh$_3$ (1.70 g, 6.48 mmol) in toluene (35 mL). The reaction mixture was stirred slowly at RT for 2 h to yield a green precipitate. Complex **8** was obtained as a green solid by filtration and washing with Et$_2$O (3 × 40 mL). Yield: 1.28 g, 71%.

**Computational details**. To obtain the ASE values and elucidate the mechanism, we used the original structures without changing the PPh$_3$ groups to PH$_3$ groups to obtain a more reasonable estimation of the energy differences. All of these structures evaluated were optimized at the B3LYP/6–31G* level of DFT[62–65] with an SDD basis set[66] to describe P, Cl, and Os atoms; single-point energy calculations were then performed on the mechanism using the TPSS/6-31G* method[67] with the PCM solvation method in dichloromethane[68]. Because we used simplified structures in the other calculations, all of these structures were studied using a larger basis set. The B3LYP/6-311++G** level of DFT[62–65] with a double-$\zeta$ valence basis set (LanL2DZ) was used to describe the P, Cl, and Os atoms. Frequency calculations were performed to confirm the characteristics of all the calculated structures as minima. In all calculations, the effective core potentials (ECPs) reported by Hay and Wadt with polarization functions were added for P ($\zeta$(d) = 0.34), Cl ($\zeta$(d) = 0.514), Os ($\zeta$(f) = 0.886)[69]. NICS values were calculated at the B3LYP-GIAO/6-311++G** level. All the above calculations were performed using the Gaussian 09 software package[70]. The AICD calculations were performed using AICD software[58].

**Data availability**. Data relating to the X-ray crystal structures are available from the Cambridge Crystallographic Data Centre (CCDC) under deposition numbers CCDC 1506325 (**1**), CCDC 1506339 (**5-PF$_6$**), CCDC 1548457 (**6**), CCDC 1548468 (**7**), and CCDC 1548472 (**9**). These data can be obtained free of charge from the Cambridge Crystallographic Date Centre via www.ccdc.cam.ac.uk/data_request/cif. For the XYZ coordinates of the optimized structures in the DFT studies of the aromaticity and mechanism, see Supplementary Data 1 and 2. The authors declare that all other data supporting the findings of this study are available within the article and its Supplementary Information file.

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

## Acknowledgements

This research was supported by the National Natural Science Foundation of China (Nos. 21490573 and 21332002) and Project funded by China Postdoctoral Science Foundation (No. 2016M602069). We thank Prof. Zhenyang Lin (Hong Kong University of Science and Technology), Prof. Qiyu Zheng (National Natural Science Foundation of China), Prof. Michael M. Haley (University of Oregon), and Prof. Roald Hoffmann (Cornell University) for their helpful advice on the preparation of this manuscript.

## Author contributions

H.X. conceived the project. Q.Z., J.L., X.Z., Y.S. and S.C. performed the experiments. Q. Z., J.L., Y.S. and S.C. recorded all NMR data. Q.Z. and Z.C. solved all X-ray structures. Q. Z., H.Z. and H.X. analysed the experimental data. H.Z. conceived the theoretical work. Y. H. and X.Z. conducted the theoretical work. J.Z., H.Z., Q.Z., Y.H., X.Z. and H.X. analysed and interpreted the computational data. H.Z., Q.Z. and H.X. drafted the paper, with support from Y.H., X.Z. and J.Z. All the authors discussed the results and contributed to the preparation of the final manuscript.

## Additional information

**Competing interests:** The authors declare no competing financial interests.

