## [Peer Review File · Nature Communications]

Reviewers' comments:

Reviewer #1 (Remarks to the Author):

The paper by Xia et al is scientifically sound and outlines a novel technique for the preparation of the metallacyclic structures. These systems are extremely interesting and raise some fundamental questions about aromaticity. The use of systems that contain multiple alkyne and allene functionalities allows for the preparation of these compounds in a single reaction event and the introduction of extra pi bonds allows additional ring fusions in the osmapentalyne/osmapentalene systems. The only negatives are that there are some situations where the English usage is "grammatically correct but awkward" and there is a general downplay of previous work in the area that needs to be brought into perspective.

1. The authors have a rich history of investigating osmapentalynes: Nature Chem. 2013 p 698, Angew. Chem. Int. Ed. 2014 p 6232, same journal 2015 p 6181, same journal 2015 p 7189, Chem. Sci. 2016 p 1815, J. Am. Chem. Soc. 2017 p 1822, Chem. Eur. J. 2017 p 6426). The osmapentalyne systems date back to 2013 and are part of a continuing research effort by these authors. The existence of prior work in this area is not mentioned in the introduction at all and does not appear until line 92. Furthermore, in most of these articles there is also a computational component probing the aromaticity of these systems. My take on this study is that the novelty of this work is that in these intramolecular systems they can achieve the core ring system in a single step from either of the two osmium complexes employed and that some related tricyclic systems could be produced from groups with additional alkyne functionality or allene functionality. This reviewer feels that prior work has not been placed into proper perspective and that a clear and strong statement that distinguishes this work from prior work, and justifies publication of a communication in a high impact journal, is lacking in this manuscript. It should also go in the introduction.

2. Isolation of 9 is nicely consistent with the DFT study of the reaction pathway since that compound corresponds to intermediate B1, which has the highest activation energy for conversion (forward or backward) amongst the intermediate compounds. The authors however have not mentioned that they have isolated analogous compounds in other studies of propargylic alcohol systems (Organometallics 2016, p 1497) lacking the second alkyne group.

3. The first two sentences (lines 24-26) make no sense. Are they trying to say that organometallic chemistry could not develop without carbon metal bonds???? Why waste journal space stating something so ridiculously obvious.

4. Abstract: Comments about atom economy are not appropriate for this type of study. A reaction that was stoichiometric in osmium could only claim atom economy if there was a practical application for the osmacyclopentalynes themselves. The originator of atom economy, Trost, is using the term to sell a process where one organic molecule transforms into another organometallic molecule using an organotransition metal catalyst. Although some impressive C-C bond formations have occurred in the studies of this manuscript, the demetallation step is lacking. Basically, the claim of atom economy in this manuscript is like claiming that the Wittig reaction is atom economical by looking only at the synthesis of the phosphonium salt.

5. The Discussion (lines 240-249) looks more like Conclusions or Summary.

Reviewer #2 (Remarks to the Author):

The manuscript entitled "Multiyne Chains Chelating Osmium via Three Metal-Carbon σ Bonds" is a very interesting and very important work. The authors developed a new way of making metal chelates with three or more metal-carbon σ bonds in a one-pot reaction under mild conditions.

They explain this reaction as an aromaticity-driven process. I believe that their theoretical and experimental arguments indeed proved that statement. This method certainly has a great potential for the preparation of polycyclic organometallic complexes. The manuscript is well written. The conclusions are supported by the experimental and theoretical data and are credible. I recommend this good work for publication in Nature Communication as is.

Reviewer #3 (Remarks to the Author):

Crystallographic reviewer.

In general, the structures have been correctly determined. However, unfortunately, each structure has a number of issues that need to be overcome.

There is some information missing from four of the structures (1, 5-PF6, 6 and 7):

_exptl_absorpt_process_details ?

_computing_cell_refinement ?

_computing_data_collection ?

_computing_data_reduction ?

_atom_sites_solution_secondary ? (missing for all structures)

This information should be supplied for all structures.

The supporting information section contains a number of errors:

"Single-crystal X-ray diffraction data were collected on an Agilent SuperNova Dual system or an Oxford Gemini S Ultra CCD Area Detector with graphite-monochromated Mo K α radiation (λ = 0.71073 Å) or Cu K α radiation (λ = 1.54178 Å)."

This sentence implies that the SuperNova system has Mo radiation, and that the Gemini system has Cu radiation, and that both are graphite-monochromated. Examining the CIF, it becomes clear that the Gemini system was the one used for the Mo-radiation collection, which is graphite-monochromated, and the SuperNova system was the Cu radiation system, and this one is mirror-monochromated. This should be clear in the supporting information.

"All data were corrected for absorption effects using the multi-scan technique."

This should be referenced, either to the program used (presumably CrysAlis Pro), and the original Blessing paper.

"The structures were solved by Patterson methods, expanded by difference Fourier syntheses and refined by full-matrix least-squares refinement of F² using the 2014 version of the program SHELXL."

SHELXL is only a refinement program. In addition, the version used for refinement was in all cases 2016/6, which should be referenced. The structures were solved with a mixture of methods and programs, which should be correctly identified and referenced:

Structure 1, solved by heavy atom methods with SIR2004

Structures 5-PF6 and 6, solved by direct methods with SIR2004

Structure 7, solved with heavy atom methods with SHELXS

Structure 9, solved with iterative method with SHELXT

The comments regarding disorder modelling will largely be addressed in comments about the individual structures, however a common problem with the structures is the use of ISOR restraints - these are intended for use with isolated atoms, not for bonded atoms. Instead RIGU, DELU, and SIMU are appropriate restraints to use in these cases. Also note that any disorder modelling should be explicitly discussed in the _refine_special_details section of the CIF for each structure.

The responses to Check CIF Reports that appear in the supplementary information should appear in the individual CIFs as _vrf responses. They will then appear in the CIF report itself.

Regarding the common problem for the structures 1, 5-PF6, 6 and 7 (missing reflections). It is not explained why these are the various values of $\theta(\text{min})$ - is this due to geometric constraints of the diffractometer, or some other reason, such as restrictions applied during data reduction?

The authors have made some efforts towards a common labelling scheme across the structures, however the rather scattergun approach to labelling carbon atoms should be addressed. It is also significantly easier for other users of the CIF if the atom lists are sensibly sorted.

Structure 1:

The main molecule has no structural problems, however there is a number of issues with the dichloromethane solvent molecules. It should be noted that the median Cl-C-Cl angle of dichloromethane from a search of the CSD (22042 hits) 112 degrees. Solvent molecules C1D and C2D are fine. Molecules C4D, C5D, and C0AA need additional restraints, and probably additional disorder modelling to achieve appropriate bond angles. Molecules C3D and C6D actually join together to form a chain (this is why hydrogen atoms could not be added). They should be investigated and the disorder more carefully modelled, included using PART and JOIN commands as appropriate.

Structure 5-PF6:

Hydrogens should be added to atom O0AA - the presence of hydrogen bonds makes these locatable. A hydrogen bonding table should be included in the CIF.

Structure 6:

The modelling of the disordered COOMe should be fixed - the positions of the C=O and C-O-Me should be separated out, which will reduce the thermal motion of that part of the molecule. In general this structure could benefit from some global thermal parameter restraints.

Structure 7:

The angle for dichloromethane C12 is significantly different from the expected value. Atom Cl4A of dichloromethane C2A should be split to give more appropriate angles for both disorder components.

Structure 9:

The explanation of the CIF check error for this structure seems unlikely. There is no mention of how the twinning is determined, and what the nature of it is. It is far more likely to be an inadequate absorption correction in the presence of a heavy atom when using Cu radiation. The instrument used (a SuperNova), by default, collects a movie of the crystal which allows a face-indexed absorption correction to be performed. This should be done.

Re: Revision Requested for Manuscript NCOMMS-17-12233

We express our sincere thanks to the reviewers for their positive and constructive comments on our work. We have taken into account all of the points raised by the reviewers and have highlighted the changes in the revised manuscript. Point-by-point responses to the issues are provided below.

Reviewer 1:

Remarks to the Author

The paper by Xia et al. is scientifically sound and outlines a novel technique for the preparation of the metallacyclic structures. These systems are extremely interesting and raise some fundamental questions about aromaticity. The use of systems that contain multiple alkyne and allene functionalities allows for the preparation of these compounds in a single reaction event and the introduction of extra pi bonds allows additional ring fusions in the osmapentalene/osmapentalene systems. The only negatives are that there are some situations where the English usage is “grammatically correct but awkward” and there is a general downplay of previous work in the area that needs to be brought into perspective.

Response: Thank you for the positive comments and valuable suggestions. We added comments on previous work in the area in the introduction and in other appropriate places. The English in the revised manuscript was modified by Nature Research Editing Service.

The authors have a rich history of investigating osmapentalynes: Nature Chem. 2013 p 698, Angew. Chem., Int. Ed. 2014 p 6232, same journal 2015 p 6181, same journal 2015 p 7189, Chem. Sci. 2016 p 1815, J. Am. Chem. Soc. 2017 p 1822, Chem. Eur. J. 2017 p 6426. The osmapentalene systems date back to 2013 and are part of a continuing research effort by these authors. The existence of prior work in this area is not mentioned in the introduction at

all and does not appear until line 92. Furthermore, in most of these articles there is also a computational component probing the aromaticity of these systems. My take on this study is that the novelty of this work is that in these intramolecular systems they can achieve the core ring system in a single step from either of the two osmium complexes employed and that some related tricyclic systems could be produced from groups with additional alkyne functionality or allene functionality. This reviewer feels that prior work has not been placed into proper perspective and that a clear and strong statement that distinguishes this work from prior work, and justifies publication of a communication in a high impact journal, is lacking in this manuscript. It should also go in the introduction.

Response: Thank you for the suggestions. As per the suggestions, we summarized our prior work in a separate paragraph in the introduction part, especially emphasizing the difference between this work and the previous work (note: Due to the limited reference number of Nature Communications, we only added the most relevant and recently published papers). Thank the reviewer for pointing out the novelty of this work. As stated in the introduction, the construction of more than two metal-carbon σ bonds from a single organic molecule and a single metal centre remains a challenge. For our previous accessed multidentate chelates, the metal-carbon σ bonds in these chelates could only be constructed via multiple steps under harsh conditions, and which required the use of a metal vinyl complex as the starting material. In sharp contrast, in this study, we designed and synthesized innovative multiyne chains that can be used to directly construct three or more metal-carbon σ bonds in a single reaction by chelating the commercially available $\text{OsCl}_2(\text{PPh}_3)_3$ or even a simple metal salt (K_2OsCl_6) under ambient conditions. This new technique both drastically simplifies the synthesis of the structures being reported and has the ability to achieve unique frameworks.

Isolation of **9** is nicely consistent with the DFT study of the reaction pathway since that compound corresponds to intermediate **B1**, which has the highest activation energy for conversion (forward or backward) amongst the intermediate compounds. The authors however have not mentioned that they have isolated analogous compounds in other studies of propargylic alcohol systems (Organometallics 2016, p 1497) lacking the second alkyne group.

Response: Thank you for your encouraging comments and constructive suggestions. We have added two examples of relevant metal vinyl intermediates isolated in other studies of propargylic alcohol systems following the introduction of 9.

The first two sentences (lines 24-26) make no sense. Are they trying to say that organometallic chemistry could not develop without carbon metal bonds???? Why waste journal space stating something so ridiculously obvious.

Response: Following this suggestion, we have revised the first two sentences.

Abstract: Comments about atom economy are not appropriate for this type of study. A reaction that was stoichiometric in osmium could only claim atom economy if there was a practical application for the osmacyclopentynes themselves. The originator of atom economy, Trost, is using the term to sell a process where one organic molecule transforms into another organometallic molecule using an organotransition metal catalyst. Although some impressive C-C bond formations have occurred in the studies of this manuscript, the demetallation step is lacking. Basically, the claim of atom economy in this manuscript is like claiming that the Wittig reaction is atom economical by looking only at the synthesis of the phosphonium salt.

Response: Thank you for the suggestions. We have modified the atom economy comments in the revised manuscript.

The Discussion (lines 240-249) looks more like Conclusions or Summary.

Response: Thank you for your comments. During the preparation of our manuscript, we read the “Guide to Authors” of Nature Communications carefully and referred to recent articles in the area of chemistry published in Nature Communications. The “Guide to Authors” states “The main text of an Article should begin with an introduction, followed by sections headed Results, Discussion (if appropriate) and Methods (if appropriate)”, where there is no strict limitation on the content of each section of the article. Of the recently reported chemical-related articles in Nature Communications, a considerable proportion placed the results and discussion contents together in the Results section, and the summary or conclusion were included in the Discussion section; thus our initial manuscript took the same approach. However, as noted by the reviewer, we are aware that such an approach may confuse the readers to some extent. Thus, we relocated the content of the discussion (which mainly presents data on the mechanism of action) from the Results section to the Discussion section in the revised manuscript.

Reviewer 2

The manuscript entitled "Multiyne Chains Chelating Osmium via Three Metal-Carbon σ Bonds" is a very interesting and very important work. The authors developed a new way of making metal chelates with three or more metal-carbon σ bonds in a one-pot reaction under mild conditions. They explain this reaction as aromaticity driven process. I believe that their theoretical and experimental arguments indeed proved that statement. This method certainly has a great potential for the preparation of polycyclic organometallic complexes. The manuscript is well written. The conclusions are supported by the experimental and theoretical data and credible. I recommend this good work for publication in Nature Communication as is.

Response: Thank you for your positive comments

Reviewer 3 (Crystallographic reviewer)

In general, the structures have been correctly determined. However, unfortunately, each structure has a number of issues that need to be overcome.

Response: Thank you for your constructive suggestions. We further refined the structures following these suggestions and addressed most of the issues. As some of the structures were re-solved and further refined, some crystal parameters, such as the bond lengths and bond angles, were changed. These changes are highlighted in the revised manuscript.

There is some information missing from four of the structures (**1**, **5-PF6**, **6** and **7**):

_exptl_absorpt_process_details ?

_computing_cell_refinement ?

_computing_data_collection ?

_computing_data_reduction ?

_atom_sites_solution_secondary ? (missing for all structures)

This information should be supplied for all structures.

Response: We added the missing information. Please see the revised CIFs for details.

The supporting information section contains a number of errors:

"Single-crystal X-ray diffraction data were collected on an Agilent SuperNova Dual system or an Oxford Gemini S Ultra CCD Area Detector with graphite-monochromated Mo K α radiation ($\lambda = 0.71073 \text{ \AA}$) or Cu K α radiation ($\lambda = 1.54178 \text{ \AA}$)." This sentence implies that SuperNova system has Mo radiation, and that the Gemini system has Cu radiation, and that both are graphite monochromated. Examining the CIF, it becomes clear that the Gemini system was the one used for the Mo-radiation collection, which is graphite monochromated,

and the SuperNova system was the Cu radiation system, and this one is mirror-monochromated. This should be clear in the supporting information.

Response: We have corrected it. Please see Supplementary Information in the second page for the correction.

"All data were corrected for absorption effects using the multi-scan technique."

This should be referenced, either to the program used (presumably CrysAlis Pro), and the original Blessing paper.

Response: The program used was cited in the revised version.

"The structures were solved by Patterson methods, expanded by difference Fourier syntheses and refined by full-matrix least-squares refinement of F2 using the 2014 version of the program SHELXL."

SHELXL is only a refinement program. In addition, the version used for refinement was in all cases 2016/6, which should be referenced. The structures were solved with a mixture of methods and programs, which should be correctly identified and referenced:

Structure **1**, solved by heavy atom methods with SIR2004

Structures **5-PF₆** and **6**, solved by direct methods with SIR2004

Structure **7**, solved with heavy atom methods with SHELXS

Structure **9**, solved with iterative method with SHELXT

*Response: Thank you for your comments. We re-solved and further refined the structures. **1** and **9** were solved by dual methods with SHELXT, while **5-PF₆**, **6** and **7** were solved by heavy*

atom methods with SHELXS. The methods and programs were corrected now and references were also added in the revised version.

The comments regarding disorder modelling will largely be addressed in comments about the individual structures, however a common problem with the structures is the use of ISOR restraints - these are intended for use with isolated atoms, not for bonded atoms. Instead RIGU, DELU, and SIMU are appropriate restraints to use in these cases. Also note that any disorder modelling should be explicitly discussed in the `_refine_special_details` section of the CIF for each structure.

Response: As per the valuable suggestions, we used RIGU, DELU and SIMU restraints for bonded atoms and discussed the disorder modelling in the `_refine_special_details` section of the CIF for each structure.

The responses to Check CIF Reports that appear in the supplementary information should appear in the individual CIFs as `_vrf` responses. They will then appear in the CIF report itself.

Response: The responses to Check CIF Reports have placed in the revised CIFs as `_vrf` responses.

Regarding the common problem for the structures **1**, **5-PF₆**, **6** and **7** (missing reflections). It is not explained why these are the various values of theta(min) - is this due to geometric constraints of the diffractometer, or some other reason, such as restrictions applied during data reduction?

Response: We used a dual-source diffractometer, and the beam-stop mask of the dual-source diffractometer blocked a proportion of the reflections.

The authors have made some efforts towards a common labelling scheme across the structures, however the rather scattergun approach to labelling carbon atoms should be addressed. It is also significantly easier for other users of the CIF if the atom lists are sensibly sorted.

Response: We have labelled the atoms in order and sorted the atom lists logically. Please note that some of the atom labels have changed because of the re-labelling, especially those of the solvent molecules.

Structure 1:

The main molecule has no structural problems, however there is a number of issues with the dichloromethane solvent molecules. It should be noted that the median Cl-C-Cl angle of dichloromethane from a search of the CSD (22042 hits) 112 degrees. Solvent molecules C1D and C2D are fine. Molecules C4D, C5D, and C0AA need additional restraints, and probably additional disorder modelling to achieve appropriate bond angles. Molecules C3D and C6D actually join together to form a chain (this is why hydrogen atoms could not be added). The should be investigated and the disorder more carefully modelled, included using PART and JOIN commands as appropriate.

Response: Thank you for your insightful suggestions. We have added additional restraints to molecules C4D, C5D, and C0AA, which now correspond to C6S, C8S and C7S, respectively, achieving appropriate bond angles. Molecules C3D and C6D have been separated using the PART command, and C3D was split into C4S and C5S in a more appropriate manner. However, due to the disorder of C6D, now corresponding to C2S, hydrogen atoms still cannot be added.

Structure 5-PF₆:

Hydrogens should be added to atom O0AA - the presence of hydrogen bonds makes these locatable. A hydrogen bonding table should be included in the CIF.

Response: Hydrogens have been added to atom O0AA which now is O1S. The hydrogen bonding table has also been included in the CIF.

Structure 6:

The modelling of the disordered COOMe should be fixed - the positions of the C=O and C-O-Me should be separated out, which will reduce the thermal motion of that part of the molecule. In general this structure could benefit from some global thermal parameter restraints.

Response: The modelling of the disordered COOMe was fixed and the positions of the C=O and C-O-Me were also separated out.

Structure 7:

The angle for dichloromethane C12 is significantly different from the expected value. Atom Cl4A of dichloromethane C2A should be split to give more appropriate angles for both disorder components.

Response: The angle for dichloromethane C12 (now corresponding to C3S) has been restrained to reasonable values using the DFIX command and atom Cl4A was also split to Cl8S and Cl10.

Structure 9:

The explanation of the CIF check error for this structure seems unlikely. There is no mention of how the twinning is determined, and what the nature of it is. It is far more likely to be an inadequate absorption correction in the presence of a heavy atom when using Cu radiation. The instrument used (a SuperNova), by default, collects a movie of the crystal which allows a face-indexed absorption correction to be performed. This should be done.

*Response: Thank you for your valuable suggestion. Considering the presence of a very heavy atom (Os) in the structure, the thin plate shape of the crystal (0.2*0.1*0.02 mm³, see its photo below) and the strong Cu radiation of SuperNova, it is reasonable to assume that the residual densities around the Os atom are attributed to the absorption issues. Following the suggestion of the reviewer, we performed a face-indexed absorption correction to the crystal, but the situation did not change. This effect may be attributed to the very thin plate shape of the crystal, impeding the determination of its exact shape. We deeply appreciate your suggestion and have tried our best to resolve the error, but it still remains unsolved. However, we believe that this error does not influence the structural identification of complex **9**, which has also been fully characterized by other characterization methods, including NMR spectroscopy, high-resolution mass spectrometry and elemental analyses.*

Reviewers' comments:

Reviewer #3 (Remarks to the Author):

It is pleasing to see the additional efforts made by the authors to correct their crystal structures. Unfortunately there are still come issues with these structures.

General Issues

"Regarding the common problem for the structures 1, 5-PF6, 6 and 7 (missing reflections). It is not explained why these are the various values of theta(min) - is this due to geometric constraints of the diffractometer, or some other reason, such as restrictions applied during data reduction? Response: We used a dual-source diffractometer, and the beam-stop mask of the dual-source diffractometer blocked a proportion of the reflections."

The beamstop mask used by CrysAlisPro is relatively conservative (it covers a very large area). There are a couple of strategies to use to adjust for this. One is to reduce the beamstop dimensions in CrysAlis before data reduction – go to CMD, then "Options RED", then the "Beam stop" tab. First turn on the beamstop overlay, then adjust the Diameter parameter. It may be necessary to adjust eh X-offset and Y-offset as well to more closely align the smaller mask. Another alternative is to completely turn off the beamstop correction (select "None" for beam stop support orientation), and instead use the profile rejection function in the manual data reduction option of CrysAlisPro (i.e. use "Data reduction with options", and at Step 3, choose "Edit special pars", and turn on the "Reject reflections with bad profiles (e.g. for HP data)" option). For all structures asides from 9, the unitcell contents are outside the unit cell boundaries. This can be straightforwardly fixed using the Olex2 tool "Centre on Cell" (View -> Symmetry Generation -> Symmetry Tools).

The authors should be careful with the use of the words constraint and restraint, as they have specific meaning in crystallography. SIMU, RIGU and DFIX are all restraints – these restrain a particular value or set of values to match a target value within a certain tolerance. A constraint, on the other hand, fixes a particular value precisely to a target. Examples of constrained values are things like fixed occupancies, or coordinates of atoms on special positions – constrained values have no uncertainties associated with them, unless the target they are constrained to has uncertainties associated with it. EADP is a constraint, as it fixes the thermal parameters of particular atoms to be exactly the same. These thermal parameters will have uncertainties, but they will be identical for all the atoms constrained this way. The `_refine_special_details` sections for all structures should be rewritten to appropriately reflect the use of constraints and restraints correctly. A better way of expressing the use of SIMU is to say "Similarity restraints (SIMU) were applied to the thermal parameters of the disordered atoms."

Structure 1 issues:

"However, due to the disorder of C6D, now corresponding to C2S, hydrogen atoms still cannot be added."

The use of negative PARTs suppresses bonds to symmetry equivalents. This will enable addition of hydrogen atoms to C2S. It should also be noted that one of the chlorine atoms in the chosen asymmetric unit for this DCM is the incorrect symmetry equivalent. As DCM C7S is only $\frac{1}{4}$ occupancy, it is better in this case to refine the atoms isotropically – the extremely tight SIMU applied was essentially refining the thermal parameters as equivalent in any case. An RES file with these changes made has been attached.

As more general note for this structure regards the formula – if component is known to be missing part of it (usually hydrogen atoms on a water molecule, in this case on the DCM C2S), the missing parts should still be included in the formula – this gives the correct physical constants (Mr, density etc) calculated from the structure. In this case the authors correctly included these hydrogens in the `_chemical_formula_moiety`, but did not in the `_chemical_formula_sum`, which is what physical constants are calculated from.

Structure 5-PF6:

Hydrogen atoms for the partially occupied solvent water molecule should be included in the formula.

Structure 6:

The authors should note that EADP is a constraint, not a restraint. The target value for the disorder C=O group is too large. The CSD average is 1.202Å, with a very tight distribution. The use of EADP in this position is probably not appropriate – this is not really what this constraint is intended to be used for. If a sensible anisotropic thermal parameter model cannot be achieved (even with the use of restraints), it is perfectly acceptable to refine a partially occupied atom isotropically. The hydrogen atoms for the water molecule should be included in the formula.

Structure 7:

This structure is fine, however the hydrogen atoms for the solvent water molecule should be included in the formula.

Structure 9:

The authors should note that for very thin crystals, correctly assigning the thickness can have a significant effect on the quality of the absorption correction. It can be difficult to correctly determine this, and manually varying the crystal size in the model can result in significant differences in the quality of the absorption correction. Another tactic for dealing with situation is to manually increase the μ^*r value in the absorption correction, which can also reduce the size of the residuals encountered.

Re: Revision Requested for Manuscript NCOMMS-17-12233A

We express our sincere thanks to the reviewer for the constructive comments on our work. We have taken into account all of the points raised by the reviewer and have highlighted the changes in the revised manuscript. Point-by-point responses to the issues are provided below.

Reviewer #3

Remarks to the Author:

It is pleasing to see the additional efforts made by the authors to correct their crystal structures. Unfortunately there are still some issues with these structures.

Response: Thank you for your valuable suggestions. Data reduction (in some case even data collection) and structure refinement were further conducted following the suggestions and all of the issues were addressed or improved. The changed crystal parameters, such as bond lengths and bond angles, were highlighted in the revised manuscript.

General Issues

"Regarding the common problem for the structures **1**, **5-PF₆**, **6** and **7** (missing reflections). It is not explained why these are the various values of theta(min) - is this due to geometric constraints of the diffractometer, or some other reason, such as restrictions applied during data reduction?"

Response: We used a dual-source diffractometer, and the beam-stop mask of the dual-source diffractometer blocked a proportion of the reflections."

The beamstop mask used by CrysAlisPro is relatively conservative (it covers a very large area). There are a couple of strategies to use to adjust for this. One is to reduce the beamstop dimensions in CrysAlis before data reduction – go to CMD, then “Options RED”, then the

“Beam stop” tab. First turn on the beamstop overlay, then adjust the Diameter parameter. It may be necessary to adjust the X-offset and Y-offset as well to more closely align the smaller mask. Another alternative is to completely turn off the beamstop correction (select “None” for beam stop support orientation), and instead use the profile rejection function in the manual data reduction option of CrysAlisPro (i.e. use “Data reduction with options”, and at Step 3, choose “Edit special pars”, and turn on the “Reject reflections with bad profiles (e.g. for HP data)” option).

*Response: Thank you for your constructive suggestions. The common problems for the structures **1**, **5-PF₆**, **6** and **7** (missing reflections) have been solved by using the first strategy suggested above.*

For all structures besides from **9**, the unit cell contents are outside the unit cell boundaries. This can be straightforwardly fixed using the Olex2 tool “Centre on Cell” (View -> Symmetry Generation -> Symmetry Tools).

Response: The unit cell contents were fixed within the unit cell boundaries following the suggestion.

The authors should be careful with the use of the words constraint and restraint, as they have specific meaning in crystallography. SIMU, RIGU and DFIX are all restraints – these restrain a particular value or set of values to match a target value within a certain tolerance. A constraint, on the other hand, fixes a particular value precisely to a target. Examples of constrained values are things like fixed occupancies, or coordinates of atoms on special positions – constrained values have no uncertainties associated with them, unless the target they are constrained to has uncertainties associated with it. EADP is a constraint, as it fixes the thermal parameters of particular atoms to be exactly the same. These thermal parameters will have uncertainties, but they will be identical for all the atoms constrained this way. The `_refine_special_details` sections for all structures should be rewritten to appropriately reflect the use of constraints and restraints correctly. A better way of expressing the use of SIMU is

to say “Similarity restraints (SIMU) were applied to the thermal parameters of the disordered atoms.”

Response: Thank you for your comments. We rewrote the `_refine_special_details` sections for all structures following the suggestions. Please see the revised CIFs for details.

Structure **1** issues:

“However, due to the disorder of C6D, now corresponding to C2S, hydrogen atoms still cannot be added.”

The use of negative PARTs suppresses bonds to symmetry equivalents. This will enable addition of hydrogen atoms to C2S. It should also be noted that one of the chlorine atoms in the chosen asymmetric unit for this DCM is the incorrect symmetry equivalent. As DCM C7S is only ¼ occupancy, it is better in this case to refine the atoms isotropically – the extremely tight SIMU applied was essentially refining the thermal parameters as equivalent in any case. An RES file with these changes made has been attached.

As more general note for this structure regards the formula – if component is known to be missing part of it (usually hydrogen atoms on a water molecule, in this case on the DCM C2S), the missing parts should still be included in the formula – this gives the correct physical constants (Mr, density etc) calculated from the structure. In this case the authors correctly included these hydrogens in the `_chemical_formula_moiety`, but did not in the `_chemical_formula_sum`, which is what physical constants are calculated from.

*Response: We re-collected the crystal data of complex **1** and the above-mentioned issues derived from the severely disordered solvent molecules were addressed successfully. The missing component (hydrogen atoms on partially occupied water molecules for the re-cultured crystal) was also included in the formula.*

Structure **5-PF₆**:

Hydrogen atoms for the partially occupied solvent water molecule should be included in the formula.

Response: Hydrogen atoms have been included in the formula in the revised CIF.

Structure 6:

The authors should note that EADP is a constraint, not a restraint. The target value for the disorder C=O group is too large. The CSD average is 1.202Å, with a very tight distribution. The use of EADP in this position is probably not appropriate – this is not really what this constraint is intended to be used for. If a sensible anisotropic thermal parameter model cannot be achieved (even with the use of restraints), it is perfectly acceptable to refine a partially occupied atom isotropically. The hydrogen atoms for the water molecule should be included in the formula.

Response: As per the valuable suggestions, the disorder atoms of COOMe group were refined isotropically and the hydrogen atoms for the water molecule were also included in the formula.

Structure 7:

This structure is fine, however the hydrogen atoms for the solvent water molecule should be included in the formula.

Response: The hydrogen atoms for the water molecule have been included in the formula.

Structure 9:

The authors should note that for very thin crystals, correctly assigning the thickness can have a significant effect on the quality of the absorption correction. It can be difficult to correctly determine this, and manually varying the crystal size in the model can result in significant differences in the quality of the absorption correction. Another tactic for dealing with situation is to manually increase the μ^*r value in the absorption correction, which can also reduce the size of the residuals encountered.

Response: Thank you for your valuable suggestion. By checking the reference frames of crystal 9, we noticed that the crystal orientation changed during the testing process, so we did data reduction again by choosing the “Follow sudden (discontinuous) changes of sample orientation” option in the “Basic algorithm parameters” step, and now the size of the residuals around the osmium center reduced significantly. After that, we tried both of the suggested methods, but the situation didn’t further improve. In addition, we also tried to resolve the issue completely by re-collecting crystal data. However, due to the unstable nature of intermediate 9, all of the crystals obtained were not ideal enough to get a better result than the original data of crystal 9. We deeply appreciate your suggestion and have tried our best to resolve the issue. We believe that the issue now does not influence the structural identification of complex 9, which has also been confirmed by NMR spectroscopy, high-resolution mass spectrometry and elemental analyses.

REVIEWERS' COMMENTS:

Reviewer #3 (Remarks to the Author):

I now consider these crystal structure suitable for publication. I am impressed by the authors diligent efforts to alleviate the issues with their crystal structures.